# Why Do We Not Follow Lifesaving Rules? Factors Affecting Nonadherence to COVID-19 Prevention Guidelines in Indonesia: Healthcare Professionals’ Perspectives

**DOI:** 10.3390/ijerph19148502

**Published:** 2022-07-12

**Authors:** Nelsensius Klau Fauk, Alfonsa Liquory Seran, Christopher Raymond, Maria Silvia Merry, Roheena Tahir, Gregorius Abanit Asa, Paul Russell Ward

**Affiliations:** 1Research Centre for Public Health Policy, Torrens University Australia, Adelaide 5000, Australia; nelsensius.fauk@torrens.edu.au (N.K.F.); christopher.raymond@student.torrens.edu.au (C.R.); gregorius.asa@student.torrens.edu.au (G.A.A.); 2Institute of Resource Governance and Social Change, Kupang 85227, Indonesia; 3Atapupu Public Health Centre, Health Department of Belu District, Atambua 85752, Indonesia; liquory.seran@yahoo.com; 4Medicine Faculty, Duta Wacana Christian University, Yogyakarta 55224, Indonesia; silvi.tropmed@yahoo.com; 5College of Medicine and Public Health, Flinders University, Adelaide 5042, Australia; roheena.tahir@flinders.edu.au

**Keywords:** perspectives, socio-cultural barriers, attitudes, subjective norms, perceived behavioural control, healthcare professionals, community members, COVID-19 prevention guidelines, qualitative study, Indonesia

## Abstract

This study aimed to understand Indonesian healthcare professionals’ (HCPs) perceptions and experiences regarding barriers to both HCP and community adherence to COVID-19 prevention guidelines in their social life. This methodologically qualitative study employed in-depth interviewing as its method for primary data collection. Twenty-three HCP participants were recruited using the snowball sampling technique. Data analysis was guided by the Five Steps of Qualitative Data Analysis introduced through Ritchie and Spencer’s Framework Analysis. The Theory of Planned Behaviour was used to guide study conceptualisation, data analysis and discussions of the findings. Results demonstrated that HCP adherence to COVID-19 prevention guidelines was influenced by subjective norms, such as social influence and disapproval towards preventive behaviours, and perceived behavioural control or external factors. Findings also demonstrated that HCPs perceived that community nonadherence to preventive guidelines was influenced by their behavioural intentions and attitudes, such as disbelief in COVID-19-related information provided by the government, distrust in HCPs, and belief in traditional ritual practices to ward off misfortune. Subjective norms, including negative social pressure and concerns of social rejection, and perceived behavioural control reflected in lack of personal protective equipment and poverty, were also barriers to community adherence. The findings indicate that policymakers in remote, multicultural locales in Indonesia such as East Nusa Tenggara (Nusa Tenggara Timur or NTT) must take into consideration that familial and traditional (social) ties and bonds override individual agency where personal action is strongly guided by long-held social norms. Thus, while agency-focused preventive policies which encourage individual actions (hand washing, mask wearing) are essential, in NTT they must be augmented by social change, advocating with trusted traditional (adat) and religious leaders to revise norms in the context of a highly transmissible pandemic virus. Future large-scale studies are recommended to explore the influence of socio-cultural barriers to HCP and community adherence to preventive guidelines, which can better inform health policy and practice.

## 1. Introduction

Since the novel Coronavirus (SARS-CoV-2 which causes COVID-19 disease) was declared a global pandemic by the World Health Organisation (WHO) in 2020 [1], preventive guidelines aimed at individual behaviours including the wearing of facemasks, handwashing, social/physical distancing, avoiding crowded spaces, and using hand sanitiser, have been recommended by WHO and global health authorities as important strategies to reduce its transmission [2,3,4,5]. In Indonesia, the first COVID-19 cases were reported in March 2020, with subsequent rapid spread of the virus reported consistently throughout the country up until this study was conducted [6,7]. In response to this situation, the government of Indonesia has enacted national prevention guidelines, as well as large-scale social restrictions on community activities (referred to as Pembatasan Sosial Berskala Besar, or PSBB) and lockdowns to slow down or prevent massive, uncontrolled transmission of the virus [7,8].

Despite comprehensive legal and regulatory frameworks developed and instituted, widespread enforcement of COVID-19 preventive behavioural guidelines has been generally suboptimal. Research exploring healthcare professionals’ (HCPs) behaviours during the pandemic has reported a range of factors that influence variability in adherence to COVID-19 prevention guidelines. On an individual level, personal negligence, ignorance (of guidelines), and lack of commitment to COVID-19 policies and procedures have been reported as barriers to adopting preventive behaviours per recommended guidelines, occasionally resulting in HCP COVID-19 infection [9,10,11,12]. Institutional level barriers to HCP adherence to preventive guidelines include a lack of appropriate policy or guideline for controlling COVID-19 cases in healthcare settings, and unclear, inconsistent and changing guidelines to which HCPs are not properly informed and oriented [9,11,12,13,14]. Material factors influencing nonadherence of HCPs include a limited supply of personal protective equipment (PPE), hand sanitisers, and other supplies used in the prevention of infection transmission during the COVID-19 pandemic [9,11,12,14,15,16,17,18,19,20,21]. Limitations in infrastructural resources resulting in few hand washing stations and patient control or quarantine areas in emergency or other healthcare environments promoted increased risk of COVID-19 transmission by preventing sufficient adherence to prevention strategies [9,11,12,13,14]. Several studies have also reported adherence barriers arising from inadequate in-service training on COVID-19 infection prevention and control measures, the proper use of PPEs, and pitfalls and challenges in extended use of uncomfortable PPEs in patient care settings [9,11,12,14,16,17,21,22,23].

The highly decentralised and locally managed regulatory environment of the Indonesian COVID-19 response requires unique, tailor-made, and evidence-based approaches regarding community outreach, preventive strategies, and management of COVID-19 policy and guidelines. As there are limitations in the scholarly literature on the sociocultural barriers and constraints of HCP and community adherence to preventive guidelines globally and in the multicultural context of Eastern Indonesia, coupled with limitations in outreach, communication, and enforcement resources, we have chosen to investigate perspectives of HCPs about constraints in preventing behaviours among them and community members in their social life in Belu and Malaka districts, East Nusa Tenggara (Nusa Tenggara Timur or NTT), Indonesia. While an individual’s behaviour based on personal agency is not the sole determinant of pandemic outcomes, positive or negative, the social disruptions seen during COVID-19 necessitate a full analysis of individual, social, and structural determinants of health outcomes. This study contributes to improving our understanding of sociocultural barriers and the individual psychological context of pandemic-prevention behaviours as important components of a comprehensive, resilient strategy for future pandemic control. Understanding barriers and constraints of adherence is useful to inform the development of policies and intervention programs to support people’s adherence to the guidelines and improve COVID-19-related healthcare service delivery.

## 2. Methods

### 2.1. Theoretical Framework

This study used the Theory of Planned Behaviour (TPB) [24] to guide conceptualisation, data analysis, and discussion of findings with the overarching goal to deconstruct the context of how decisions were made on an individual level regarding COVID-19 prevention behaviours. The TPB framework enabled the study team to identify the psychological contexts which gave rise to nonadherence among HCPs and community members as reported by HCP respondents. Adding to the foundations of the Theory of Reasoned Action, the TPB posits several psychological domains which influence behavioural performance, specifically: behavioural intention, attitude, subjective norms and perceived behavioural control, which influence behavioural performance. When taken together, these domains comprise an individual’s ‘belief structure’ and are predictive based on how they are expressed in an individual [24,25].

These domains informed the design of the interview guides, including individual attitude as a function of perceived outcomes and efficacy of a given preventive behaviour (e.g., the utility of mask wearing to prevent transmission; positive or negative perceptions of social distancing requirements, etc.). Table 1 below outlines the linkages between TPB domains and interview questions developed in this research. Development of the research focus was predicated on the assumptions arising from TPB which suggested that these attitudes were coloured and influenced by evolving subjective and social norms as the community responded and adapted to the COVID-19 pandemic. These include normative behaviours around revised social mores and relationships, heavily depending on social and peer pressures, concerns of social rejection for noncompliance with revised social norms, and how peers approved or disapproved of behaviours specific to COVID-19 prevention guidelines (wearing masks, keeping physical distance) [24,26]. Perceived behavioural control includes those exogenous pressures and structural influences (constraints and facilitators) which exert effects on an individual and determine their perception of personal agency to act [24,26].

### 2.2. Study Setting

The study was conducted in Belu and Malaka districts, NTT, Indonesia from February to March 2022. NTT is a province located in Eastern Indonesia which shares its border with the nation of Timor Leste [27]. The archipelagic province extends over a geographic area of 47,932 km^2^ spread across 566 islands, the largest of which are Flores, Sumba, and West Timor. NTT encompasses 72 regional languages, with a sociocultural mix of Christian, Catholic, Muslim, and traditional (animist or syncretic) religions and active tribal affiliations [27]. It is among the poorest of Indonesian provinces, and thus is an important laboratory for studying the effects of the pandemic in resource-constrained settings.

NTT’s decentralised health system is characterised by a mix of public and private sector providers, including primary, public health centres, clinics, and hospitals that provide COVID-19 testing, hospitalisation and treatment, and vaccination services. NTT’s health infrastructure is limited in both human and material resources, impacting overall health outcomes, and was massively constrained in its ability to respond to the COVID-19 pandemic. Belu and Malaka districts are two of 22 districts in NTT [28], and Malaka district was part of Belu before the separation in 2012 [29,30]. Due to limited COVID-trained healthcare professionals and medical devices and equipment, only two hospitals (one public and one private hospital) in Belu and one hospital in Malaka were able to provide care and treatment for COVID-19 patients. As a consequence, these healthcare facilities were overloaded with more than 2000 COVID-19 patients during the pandemic as per the current report on 20 May 2022 [31]. As of May 2022, 93,808 people tested positive for COVID-19 in NTT, of whom 92,266 have fully recovered, with 25 hospitalisations and 1517 deaths [32].

### 2.3. Study Design 

This study employed a qualitative methodology, using in-depth one-on-one interviews guided by a questionnaire developed by the research team, investigating the subjective experience, perceptions and opinions of HCPs in the study settings. This study explored the dynamics of HCP perspectives and experiences regarding reasons underlying perceived nonadherence to preventive guidelines, to better understand how HCPs positioned themselves in providing COVID-19 health services in the community [33,34]. The research team focused on open-ended questionnaires, encouraging participants to explore ideas and to better reflect the lived reality of daily life during the COVID-19 crisis [34,35].

The aim of this study was to investigate the challenges to adherence to COVID-19 preventive guidelines and practices by exploring the ecosystem of sociocultural experience and perceptions of HCPs working in the provision of healthcare in Belu and Malaka districts, NTT. To achieve this aim, the researchers employed two research questions: (1) What are the sociocultural factors affecting HCP adherence to prevention practices in their social life, and (2) how do HCPs perceive pandemic preventive practices or nonadherence among the community members in their daily life? Investigation of individual behaviours coupled with perceived barriers among the community they serve provides unique insight into challenges in the design and implementation of official guidelines, identifies constraints among professionals and the lay community during a time of heightened urgency, and provides details on the complex social relations between healthcare providers and the public during the COVID-19 pandemic in these resource-limited settings.

These included HCPs’ views or perspectives on knowledge and understanding of community members about COVID-19-related information; their perspectives about community members’ attitudes towards COVID-19-related information disseminated by HCPs and the governments; their perspectives on how community and family members influenced each other towards COVID-19 prevention guidelines, and the influence of cultural practices of their adherence to the guidelines. The interviews also explored HCPs’ social experiences regarding COVID-19 prevention guidelines; the influence of noncompliance attitudes and behaviours of their family members and other community members on their adherence to the prevention guidelines; and the influence of family and social events on their adherence to the guidelines. To the best of our knowledge there have never been studies on this topic in the context of NTT, and due to feasibility, familiarity and the potential of undertaking the study successfully, we selected Belu and Malaka districts as the study settings.

### 2.4. Data Collection and Analysis

Twenty-three healthcare professionals (medical doctors, nurses, and pharmacists) working in government public health centres or public and private hospitals participated in this study. They were between 22 and 38 years old (See Table 2). The majority (*n* = 16) worked at community health centres, with the remaining seven employed at two public and one private hospitals in the districts. All participants were from the local communities in Belu and Malaka. Participants were recruited using the snowball sampling technique [36], by circulating the Study Information Sheet (SIS) among social and professional networks of local healthcare professionals with an invitation for application to the study. Individuals who contacted the study team were followed up directly for potential recruitment. Initial respondents were subsequently requested to distribute the SIS among their professional networks, thus snowballing the pool of potential eligible participants. For the de-identification purpose, each participant was assigned a specific study identification letter and number. Prior to conducting interviews, each participant was once again advised regarding the study purpose, the voluntary nature of their participation, their rights to withdraw participation during or after the interview, and assured that the data provided are anonymous and confidential in perpetuity. They were also advised that ethics approval for this study was obtained from the Health Research Ethics Committee, Duta Wacana Christian University, Indonesia (No. 1380/C.16/FK/2022). Participants were recruited based on several inclusion criteria, including one had to be 18 years old or above, a healthcare professional and willing to voluntarily participate in the study.

Data collection was performed using the one-on-one, in-depth interview method. Interviews were conducted either via encrypted WhatsApp video teleconferencing or through in-person, face-to-face meetings, which were decided based on the expressed preferences of the participants. Time and place for interviews were agreed upon by the field researcher and each participant, resulting in sixteen in-person interviews and seven digital video interviews. Face-to-face interviews were carried out in a private room at the participants’ place of employment during working hours. Both interviewer and participants abode by COVID-19 guidelines by maintaining a physical distance of 1.5 metres and wearing facemasks during the interviews. The seven video teleconferencing interviews were conducted after working hours or on the weekends. Only the researcher and each participant were present for the interviews in all cases, and the interviews lasted roughly 30 to 45 min. Interviews were conducted in Bahasa Indonesia, and were digitally audio recorded, with occasional supplemental notes jotted down by the researcher during the discussions. All participants who had initially responded to an expression of interest using the snowball sampling technique took part in the study, with no subsequent withdrawals, and each participant was interviewed one time only. All participants were required to read, understand, and sign the information sheet with informed consent regarding the study details, objectives, and use of data prior to their participation. 

Audio recordings were transcribed verbatim, and transcripts were thematically analysed according to the Five Steps of qualitative data analysis described in Richie and Spencer’s analysis framework [35,37]. These steps are: (1) familiarisation with the data or transcripts by repeated readings, breaking down the data into chunks through labelling and comments; (2) identification of a thematic framework by writing down key recurrent issues, concepts, and ideas; (3) indexing the dataset by creating data codes for each transcript and then listing all the codes (open coding), followed by removal or combining of redundant or repetitive codes which were thematically and sub-thematically grouped; (4) charting the data by reorganising themes and codes created in the previous steps in a summary of chart for comparison of the data within and across interviews; and (5) mapping and interpretation of the dataset as a whole once organised thematically [37,38].

## 3. Results

Interview data from this study are presented via grouping of themes generated during analysis within the framework of the TPB which reveals subjective experience of health restriction adherence during a time of heightened social pressure and urgency. Following are excerpts of the qualitative dataset which informed conclusions on the marked influence of a range of factors on community members and individual HCP adherence to COVID-19 preventions guidelines.

### 3.1. Barriers to HCP Adherence to COVID-19 Prevention Guidelines

#### 3.1.1. Subjective Norms: Social Influence and Disapproval towards Preventive Behaviours

Subjective norms, reflected in strong social influence and overt disapproval of others’ COVID-19 preventive behaviours, were a strong predictor for HCP nonadherence. Such influence and disapproval seemed to put pressure and raise concern of rejection among the participants, and lead to fear of reprisal or social ostracisation, especially in the context of family. This is supported by the following narrative of a 24-year-old nurse who described how her family members specifically asked her not to wear a facemask during a family gathering, and how she thus felt uncomfortable wearing a facemask during this social engagement:
*“Not long ago my aunty passed away, and all our family members went to her house to help. On one occasion, we were working in the kitchen, cooking and preparing food and drinks for guests, and one of my older relatives said ‘we are all family members, you do not have to wear a mask’. I felt a bit uncomfortable because of that and also because all of them did not wear a mask or maintain physical distancing, so finally, I took off mine ….”.*(P1, nurse)

Nonadherence of other community members towards preventive guidelines was a form of negative subjective norm that influenced participants’ behaviours. This resulted in a heightened concern of social rejection expressed by participants, including the potential for stigmatisation resulting from ‘following the rules’ which were deemed spurious or unnecessary by the community at large. Policies enacted to regulate based on assumptions of high individual agency did not address the psychosocial barriers when facing potential social rejection from seemingly aberrant, non-normative behaviour (forcing people to stay at a distance, covering their face, etc.), as illustrated by a hospital-based nurse expressing internal conflict over the decision between ‘adherence to guidelines’ vs. ‘social cohesion and compliance with social norms’:
*“My own experiences show that it is a bit difficult for me to fully adhere to COVID-19 prevention protocols because my neighbours and other people in the community where I live do not comply with the protocols. Whenever I meet my friends or neighbours, I always try to keep a distance, but when the conversation starts, they get closer and closer to me. This is perfectly normal, and is how we interacted with each other prior to the COVID outbreak. Although I’m aware that it is not allowed by COVID guidelines, I cannot say ‘please keep your distance from me’. I think it is unethical to do so. It can make them feel offended and they will think I am arrogant and may then avoid me”.*(P4, nurse)

Similar comments were raised by another nurse who worked for a public health centre. The participant described how difficult it was for him to avoid crowds at social events, such as birthdays, church first communions, or wedding parties. He described that it was impossible to make excuses every time he was invited by others in order to avoid the crowds, and thus succumbed to social pressures:
*“Just this month I’ve received five invitations for birthday parties and first communion celebrations in the community. As we all live together, if you’re invited by a friend or neighbour then you must attend. These are people who know you, so if you don’t attend, then what they would say about you? There were so many other people who were also invited to these parties, so it was difficult to avoid the crowds and keep physical distance from each other…these parties were not ‘adjusted’ to comply with COVID-19 prevention guidelines. These were the situations I’ve faced so far. I wore facemasks, but could not avoid the crowds or keep my distance from others as they seemed unaware or did not really care about those guidelines”.*(P11, nurse)

#### 3.1.2. Perceived Behavioural Control: External Situations

Participants also raised comments on external factors or situations as influencing their adherence to preventive guidelines. Echoing other responses, some participants expressed their personal experiences with situations which were out of their control and that influenced their adherence to preventive behaviours, as in the following from a physician:
*“There have been situations where it was impossible for me to fully comply with the COVID-19 guidelines…. Once one of my nieces got married, all of our family members had to work together and serve the wedding guests. It was impossible to keep physically distant from each other, even though I was fully aware of the prevention guidelines. The situation made it impossible for us to keep distance because we had to work together”.*(P19, medical doctor)

Additionally, another health professional remarked on the physical challenges of adherence due to external factors such as discomfort from high temperatures:
*“I wore a mask sometimes (during family and social events), but I didn’t feel okay wearing the mask for the whole day…you know, the weather was hot which made it even harder for me and everyone else to wear a mask all the time. ….”.*(P20, pharmacist)

### 3.2. Barriers to Community Member Adherence to COVID-19 Preventive Guidelines

#### 3.2.1. Behavioural Intention and Attitude

A key theme related to HCPs’ disbelief in COVID-19 information. The data generated from this study revealed that HCP respondents situated themselves as knowledge keepers of a biomedical truth of COVID-19, voicing perspectives that members of the community were mistrusting of the COVID-19 narrative as circulated by health authorities. In this specific set of interviews, a tense dynamic was revealed in the perspectives of HCPs which pinpoints blame for nonadherence on individuals’ disbeliefs, mistrust, and lack of confidence in ‘authoritative knowledge’. This assignment of agency for the community at large stands in contrast to the expressed reasons for their own (HCP) nonadherence to COVID-19 protocols, vis-à-vis the overwhelming social pressures for conformity to ‘pre- or non-pandemic’ norms. Illuminating more as viewpoints of the relationship between HCPs and the community they served, opinions on ‘why the community does not follow the rules’ are instructive:
*“There are many community members who do not believe in the existence of COVID-19 and think that it’s made up by the government. They argue that COVID has been reported everywhere, but many people have not been infected even though they gather with others and don’t keep physical distancing nor wear facemasks at social events or parties. They don’t believe in the information disseminated by the government and do not adhere to COVID-19 guidelines”.*(P5, medical doctor)

HCPs in this study generally held the view that the community lacked knowledge of COVID-19 despite having access to the specifics of government regulations. As outlined by a public health centre nurse, blame was assessed squarely on the fact that the community both lacked first-hand experience (i.e., no infections in their immediate vicinity) and had limited understanding of the relationship between recommended preventive behaviours and reducing viral transmission:
*“They (community members) do not comply with health protocols because they have very limited knowledge about COVID-19. Although the government has recommended guidelines such as wearing facemasks and keeping physical distance, they do not follow these recommendations. Some say: ‘we never use facemasks and often gather with no physical distancing but we still don’t get infected’. I think they do not get infected because they are in rural areas and hardly go out of their communities to other places; so even if they gather together, they aren’t exposed to the infection. However, if one of them were to be infected, then it would easily be transmitted among them. So, I think lack of knowledge does influence their adherence to the guidelines”.*(P9, nurse)

An additional theme related to HCPs believing that they were mistrusted within their communities. HCPs in this study expressed perceptions of mistrust on the part of the community, including remarking on conspiracy theories and deceptive practices regarding COVID-19 diagnosis in the health system. They also mentioned that patients were confused when having a positive COVID-19 test result while feeling ‘healthy’, and equated this with deceptive hospital or clinic practice with the aim of increasing revenue from government funds:
*“Distrust in healthcare professionals and hospitals about COVID-19 test results has been widespread within communities. They assume that healthcare professionals or hospitals deliberately make their test results COVID-19 positive so that the hospitals receive more funds from the national government. The higher the number of COVID-19 patients a hospital takes care of, the more the funds it will receive from the national government”.*(P17, pharmacist)

One nurse remarked on the discrepancies between ‘feeling healthy’ and being diagnosed with COVID-19, leading to widespread distrust among community members:
*“It appears that there is an opinion within communities about distrust in healthcare professionals and healthcare facilities in relation to COVID-19 test results. Such distrust seems to influence their compliance with COVID-19 prevention protocols, such as wearing facemasks and avoiding crowded places or social distancing. They do not trust the test result issued by the hospital (COVID-19 test is done only in one hospital), if it is positive, with the reason that they are not sick and feeling healthy”.*(P16, nurse)

Early on in the pandemic, confusion abounded as seemingly unrelated illnesses were diagnosed under the umbrella of COVID-19, leading to rejection by the community and mounting distrust:
*“It happened during the early period of the COVID-19 outbreak that every patient who died in the hospital is claimed by the hospital as COVID positive. That created big suspicion among patients’ families and community members. I remember, one time there was a person, an old grandpa who accidentally fell in his bathroom and was rushed to hospital. Shortly after they arrived at the hospital he passed away. Suddenly, the family members of the patient were told by a medical doctor that the patient was COVID positive. His family members did not accept and asked for proof of the test result, but the doctor failed to provide it with the reason that the test result has been thrown away. The family members got mad and beat up the doctor ……”.*(P3, nurse)

The data illustrate that HCPs have access to community rumours and explanations for COVID-19, including expressions of mis/distrust arising from confusion about the nature of viral transmission, diagnosis, and treatment. Taken at face value, these quotes underscore the perceived rift between HCPs and the community they serve, as well as delineating perceived deficiencies in access to appropriate health information and messaging.

In addition to feeling mistrusted, HCPs talked about what they saw as community beliefs in traditional ritual practices which are perceived to ward off misfortune (such as COVID-19). Traditional (adat) practices such as rituals to ward off misfortune, common throughout East Nusa Tenggara, were reported by HCPs as potentially influencing people’s adherence to COVID-19 prevention guidelines. Such practices have existed for a long time and were performed to ward off misfortunes, including various diseases before the outbreak of COVID-19. Participants described how the community believed these rituals could also protect them from COVID-19 infection. The following narrative of a medical doctor who had been part of the COVID-19 medical team since the beginning of the outbreak reflects the influences of these traditional practices on the wearing of facemasks and physical distancing among community members:
*“The traditional rituals to ward off COVID-19 are performed everywhere. I have heard of these practices a long time ago but during the COVID-19 outbreak, the rituals are held in many custom homes. You can just look at Facebook, there are many posts on these rituals. These rituals are performed to protect them from COVID-19. These are cultural practices that have been passed down from their ancestors and they believe in these rituals. The problem is that the ones who have done these rituals perceive that they are resistant to COVID-19 infection, meaning it will not infect them. They feel safe. Such perception and feeling seem to make them careless and influence their adherence to the guidelines, such as wearing facemasks and maintaining physical distance when they go out to public places”.*(P2, medical doctor)

Another participant who worked at a public health centre further commented that such rituals made community members feel safe, protected, and being ‘resistant to infection’ and led to them not wearing facemasks or maintaining physical distancing:
*“Some said ‘Ah we have done the ritual to ward off COVID-19, so we will not get infected’ even though they do not wear masks when they are in crowded places”.*(P12, nurse)

These quotations reveal that there is a strong link in this community between tradition and how they understand disease causation and prevention. The HCPs interviewed, espousing a biomedical paradigm, expressed the opinion that the ‘beliefs and attitudes’ of the community were at odds with the norms established by the healthcare system. They were abutting against longstanding tradition which informed a non-Western explanatory model for disease causation in which disease expulsion rituals were seen as efficacious responses to an unknown health threat in the community.

#### 3.2.2. Subjective Norms: Negative Social Pressure and Concerns of Social Rejection

Despite taking place during restrictions engendered by a global pandemic, social and traditional events, rituals, and gatherings continued in Belu, Malaka and NTT unabated. Social cohesion served to strengthen communities, often at the risk of increased transmission of COVID-19 when not abiding by prescribed preventive behaviours. The overwhelming drive for socialisation during a time of uncertainty in the community was reflected by a local nurse:
*“Our society is communal and has strong family and social ties. If there is an event in a community then all friends, neighbours, and other community members are invited and certainly will attend. You can imagine, ‘how people can maintain physical distancing?’…it is impossible”.*(P10, community nurse)

Adherence to long-held social norms of communality and socialising took precedence over the government restrictions around mask wearing and physical distancing, in which these ‘antisocial’ practices had not yet evolved as normative. Participants remarked that attending events such as weddings, funerals, religious rites of passage, birthday parties and others represented major challenges to community members’ ability to adhere to best practices of prevention. Despite having knowledge of COVID-19 transmission, protocols, and policies, the social pressure to conform overrode personal knowledge:
*“The practices of our social life within families and communities have a significant influence on our adherence to guidelines. Social and family gatherings for social events, such as weddings, birthday and communion parties, and funeral ceremonies, are very common. These social events involve friends, neighbours, and community members, and many do not wear facemasks or maintain physical distancing. A few people may wear facemasks, but because the majority do not wear them, they eventually take them off and put them in their pockets. Some may try to keep distance but the others do not care, so it is difficult to comply with the guidelines in such situations”.*(P6, medical doctor)

A public health centre nurse observed in her own experience that ‘familiarity’ or intimacy with eventgoers precluded illness. Familiarity was equated with purity, in the sense that known entities were uncontaminated by the COVID-19 virus:
*“I have often seen there are people who do not want to wear masks or keep social distancing in social events they attend, and even influence their friends or family members not to comply with COVID-19 guidelines. Some said to their neighbours ‘it is not necessary to wear masks. We know each other, so why we should wear masks all the time and keep distance or sit far from each other’….”.*(P15, nurse)

#### 3.2.3. Perceived Behavioural Control: External, Systemic Factors

HCPs, as members of the communities in which they worked, made frequent enquiries into the reasons behind nonadherence in day-to-day life. External constraints included lack of availability of supplies, as well as financial constraints arising from poverty. When querying the community in public venues such as markets, adherence to preventive behaviours was linked to the fact that these newly introduced commodities had not yet normalised for them:
“Sometimes I asked people at the banks, supermarkets, or public health centres about the reason why they don’t wear a facemask, and the most common answers I received were ‘I ran out of masks’ or ‘I bought masks and have used them all and forgot to buy more’. What I can say? I think people are not accustomed to providing masks at home as one of their daily needs in this COVID-19 situation”.(P8, medical doctor)

Behaviour changes in shopping habits had not yet included purchasing pandemic-prevention commodities, or that they were unavailable in public venues and seen as an afterthought:
“Some patients who were sick and came to this hospital admitted that they do not clean up their hands with hand sanitiser. They said that they do not have hand sanitiser at home or in public buildings such as stores and supermarkets they have visited. I should admit that even in many healthcare facilities hand sanitisers are not provided for visitors”.(P21, nurse)

Financial constraints arising from poverty seemed linked to community member nonadherence, both in terms of not having enough money to purchase preventive commodities, as well as forcing noncompliance with lockdowns to seek income. Closures and limitations in work opportunities during the pandemic created immense financial pressure on the community, and often they were forced to choose between income to take care of their family or abiding by social restrictions and lockdowns:
*“Poor economic or financial condition is one of the biggest influencers towards community members’ behaviours in relation to COVID-19 guidelines. Most community members rely on the sale of vegetables and fish at the market to support their daily needs. So, they go to the market every day to sell their goods, otherwise they could not buy their necessities or food to eat. They said ‘if we do not sell our goods then who will feed our families, who will provide us food? The market is a very crowded place. Many people come to the market every day, and they do not seem to really care about wearing facemasks or maintaining physical distancing”.*(P7, pharmacist)

## 4. Discussion

Guided by TPB concepts [24], our analysis demonstrates two sets of perceived standards operating in the community as reported by HCPs which led to nonadherence to COVID-19 preventive behaviours. HCPs noncompliance resulted from social pressure despite having extensive biomedical knowledge regarding the pandemic, while they reported that community members’ nonadherence resulted from social pressure because of a lack of biomedical awareness, ignorance, etc.

In this research, HCPs expressed positive attitudes, knowledge/beliefs, and intentions to adhere to established preventive protocols based on extensive biomedical, insider understanding within the health system, access to government information and epidemiology data. Given the set of structural conditions which align with an expected behavioural outcome of ‘compliance’ (mask wearing, physical distancing) on the part of HCPs, the TPB predicts that intention will match outcomes [24]. However, as demonstrated by the participants the behavioural control domain and subjective norms/normative beliefs are more strongly influenced by social pressure, the need to fit in with ‘pre-pandemic social norms’, social situations, and thus exhibit behaviour outcomes of ‘nonadherence’. 

Our findings add further evidence to the existing literature which reports attitudes, such as perceptions about the unnecessity of facemask wearing or physical distancing if having regular contact with COVID patients, and limited knowledge of how to adhere to the guidelines, as barriers to HCP adherence [14,39]. A previous study by Smith and colleagues [39] reports that subjective norms, such as greater social pressure to adopt preventive behaviours, are supporting factors for HCP adherence to preventive guidelines. Thus, it is plausible to argue that subjective norms could play a role either as facilitators or barriers to HCP adherence to COVID-19 prevention guidelines.

By contrast, the nonadherence among community members, as perceived by HCPs, was due to several factors arising from negative attitudes, beliefs, ignorance and intentions which resulted in ‘nonadherence’ to preventive practices. In the estimation of HCPs, community members’ attitudes were influenced by limited knowledge, ignorance, deliberate avoidance, and belief in superstitions or traditional cultural practices which resulted in improper behaviour. 

Numerous examples were given by HCPs for reasons influencing individual behaviour, most notably that the community expresses mistrust in health or government authorities and by extension does not trust information related to COVID-19. As per HCP narratives, the community was suspicious of the dominant biomedical explanation for the phenomenon, evidenced by the fact that many community members have no empirical experience with sick patients despite nonadherence to preventive protocols. This suspicion and mistrust of health authorities have been reported elsewhere in the scholarly literature [40,41,42,43,44,45]. HCPs reported hearing rumours and second-hand information from the community expressing incorrect and conspiratorial beliefs about the origin and nature of COVID-19. These included notions that COVID-19 was a manufactured phenomenon to manipulate the insurance reimbursement system for additional profits from testing and treatment, and that COVID-19 tests are not trustworthy and misdiagnose and amplify case numbers without (symptomatic) evidence. This aligns with other studies in Indonesia and elsewhere which demonstrated that communities with low access to COVID-19 information, or expressing beliefs of disease causation which do not align with the dominant biomedical explanations often have low levels of trust and poor adherence to pandemic restrictions [41,42,43,44,45,46,47,48,49].

In the case of NTT, this suspicion of health authorities is likely embedded in historic causes, as there has long been a struggle between the local traditional communities under various colonial entities over the past several hundred years [50,51,52,53]. A sociocultural barrier may be strongly influencing the perceptions of HCPs as to why the community does not adhere to recommended government protocols. This is especially clear in the reported use of traditional rituals to ward off misfortune (e.g., *tolak bala* rituals) that were performed by the community to remove the health threat posed by COVID-19 [51,54]. As reported elsewhere in NTT and Indonesia, these rituals served to promote social cohesion as well as to remove an entity deemed dangerous to the community [51,52,53,54]. These traditional rituals to ward off diseases align with long-held, ingrained, personal belief systems that are challenged by the rapid emergence of a Western, biomedically-dominant paradigm which challenges their usual remedy for misfortune and disease. This points to the nature of disease causation in the minds of the practitioners, which is likely a non-reductionistic, syncretic or animist ontology. Thus, in the minds of the community and in their personal experience they may indeed be afforded ‘protection’ by these rituals, and their nonadherence is logically coherent and internally consistent.

The TPB outlines the influence of ‘perceived behavioural control’ on individual behaviours; in this case, HCPs identify external factors or structural constraints such as poverty and lack of PPEs and other commodities as impacting the community adherence to preventive behaviours. Previous findings have reported that poverty due to unemployment and lack of income prevent people from purchasing PPEs and influence adherence to preventive guidelines [42,43,46,55]. Poverty also drives the need to engage socially for work which influences community adherence to preventive guidelines, such as maintaining physical distancing or avoiding gathering, as reported in previous studies [40,46]. However, based on findings from this study, community nonadherence seems highly correlated with social pressures to comply with normative behaviours in the context of ‘suspicion’ of whether COVID-19 actually exists or not based on empirical data from community members’ lived experiences. In essence, there seem to be two cultural logics or paradigms operating in parallel within the study sites: (1) a community-level empiricism that holds that COVID-19 may be fictitious, or that it can be remedied by traditional means (rituals, engaging in social cohesion activities), caused perhaps by personalistic agents (curses, sorcery, bad luck, etc.), and (2) HCPs’ empiricism based on a biomedical paradigm, COVID-19 preventive guidelines, and regulations. On the one hand, HCPs hold the community to a standard (compliance with protocols) for which they themselves are admittedly noncompliant when subjected to the same social pressures during mass gatherings, ceremonies, or other events.

Tacit assumptions made by HCPs that biomedical knowledge is authoritative and thus superior, diminished local subjectivities giving rise to sociocultural beliefs and practices. The professional distancing as expressed by HCPs underscores an ethos of the superiority of ‘expert’ vs. ‘lay knowledge’, and does not engender a sense of solidarity or empathy, and positions them as essentially victim blaming. However, in practice, the overriding values associated with subjective norms (which act as external behavioural controls) take precedence over both groups, HCPs and the community. Espousing either traditional or biomedical paradigms seemed to matter little in terms of changing behavioural outcomes: both HCPs and the public were nonadherent when faced with breaking social norms, out of fear of rejection, reprisal, or social sanctions. Cognitive dissonance on the part of HCPs is observed through a double-standard: the community is ignorant of the rules, or poor, or makes improper choices and thus do not wear masks or distance themselves at gatherings; meanwhile, HCPs are well-versed in the science and regulations around COVID-19 prevention, have access to PPEs and tools, and yet also do not wear masks or distance themselves at gatherings. These reflect different epistemologies, with the same behavioural outcomes. Within this context, both groups exhibit behaviours based on internal consistency, yet reveal that individual agency-based prevention policies do not account for variations in worldview and traditional practice, which hold sway over both community members and HCPs alike. The strong individual need for social acceptance and compliance with norms overrides perceived risk of transmission, especially given a historic context of mistrust of the community towards authorities [24,26].

Based on the four domains explored (behavioural intention, attitude, subjective norms, and perceived behavioural control), the TPB interpretation illustrates that HCPs and community members’ nonadherence stems from attitudes of mistrust and suspicion, and a need to normalise social events despite government-instituted social restrictions. Individual behaviours are highly influenced by the social milieu in NTT; and nonadherence will likely continue absent efforts to account for the overriding pressures exerted on individuals based on the socio-cultural, traditional (religious), and familial contexts. 

### Limitations and Strengths of the Study

Findings of this study should be interpreted with caution to some limitations. As is the case of many qualitative studies, this study involved a small number of participants or HCPs; thus, the current findings mainly reflect perceptions and experiences of the participants about the topics and are not meant to claim generalisability. This study did not involve community members who may have strengthened or enriched the current findings with their views and experiences on the influence of socio-cultural barriers on community adherence to preventive guidelines in the study settings. Thus, future large-scale studies are recommended to explore the influence of socio-cultural barriers to HCP and community adherence to the guidelines, which can better inform health policy and practice. The strength of this study is that it represents initial findings on socio-cultural barriers to HCP and community adherence to COVID-19 prevention guidelines, which are still limited in the current literature globally and especially in the Eastern part of Indonesia. Thus, the current findings are important and can inform the local governments of Belu, Malaka and NTT to make local adjustment to the national COVID-19 prevention guidelines in their responses to the pandemic.

## 5. Conclusions

The COVID-19 pandemic did not occur in a social vacuum; as it made its way through Indonesia, it encountered traditions, cultures, and preconceptions influencing how disease entities impact communities and health systems. This study confirmed that personal behavioural outcomes are influenced by intention, attitude, subjective norms, and perceived behavioural control, and that the various confluence of these factors can determine how an individual will behave. Structural constraints such as poverty and lack of material resources had an impact on adherence to preventive behaviours, as reported elsewhere in the literature. The strongest determinant for nonadherence, as reported by HCPs, was the need for compliance with subjective sociocultural norms and practices in family and social events, in public, and at gatherings. The pressure to appear ‘normal’ and reject the use of masks or physical distancing was more influential than personal knowledge, access to resources, or others.

The findings indicate that policymakers in remote, multicultural locales in Indonesia such as NTT must take into consideration that familial and traditional (social) ties and bonds override individual agency where personal action is strongly guided by long-held social norms. The impetus for belonging and acceptance, and to maintain psychological wellbeing is inextricably linked to social performance, cohesion, and participation in community-endorsed rituals and activities. Thus, while agency-focused preventive policies which encourage individual actions (hand washing and mask wearing) are essential, in NTT they must be augmented by social change, advocating with trusted traditional (adat) and religious leaders to revise norms in the context of a highly transmissible pandemic virus. This study demonstrates that trust and social cohesion are the primary drivers and determinants for adherence to individual prevention, and must be considered in future policy design.

## Figures and Tables

**Table 1 ijerph-19-08502-t001:** Main interview guide questions based on TPB domains.

TPB Domains	Definition	Main Interview Guide Questions
Behavioural intention	Motivational factors that influence an individual to perform a given behaviour (COVID-19 prevention guidelines)	- What motivate and demotivate you to adhere to COVID-19 prevention guidelines in your social life? Please explain. - What do you know about factors that motivate or demotivate community members’ adherence to COVID-19 prevention guidelines? Please explain.
Attitude	An individual’s belief about the outcome of performing a recommended behaviour (behavioural outcome)	- What are your perspectives on and personal experiences of COVID-19 prevention guidelines? - What are your perspectives about the level of COVID-19 knowledge, access to information, and understanding that people in your community have? - What are the attitudes of other people in your community about how the government and healthcare workers are sharing COVID-19 information? How do their attitudes influence their behaviour regarding COVID-19?
Subjective norm	An individual’s belief about whether people around them, such as within families or communities, approve or disapprove of their performance of a recommended behaviour	- How does your family or your local community influence the ways that you personally respond to the COVID-19 restrictions? Please explain more about this. - When your family and friends are not following COVID-19 restrictions, how does this make you feel? Does it change the way you behave regarding these restrictions? Please explain how this affects you. - Do you feel personally pressured to act in certain ways regarding COVID-19 restrictions based on how others around you are acting? How can you describe this pressure to act or behave in certain ways? - What are your perspectives on social pressure and influence on the adherence of community members towards COVID-19 prevention guidelines? - Are there traditions or local practices which affect how other community members are responding and behaving regarding COVID-19 restrictions? Please explain more on these. - What are your perspectives about the influence of family and social events on community members’ adherence to COVID-19 prevention guidelines?
Perceived behavioural control	External factors that may facilitate or hinder people’s ability or intention to perform a recommended behaviour	- What are factors or situations that you think influence your adherence to COVID-19 guidelines? Please explain further. - What is your view on availability of supplies, financial constraints, etc and their influence on community members’ adherence to COVID-19 prevention guidelines?

**Table 2 ijerph-19-08502-t002:** Sociodemographic profile of the participants.

Characteristics	Malaka (*n* = 10)	Belu (*n* = 13)
**Age**		
21–30	2	6
31–40	8	7
**Religion**		
Catholic	8	9
Protestant	2	4
**Education**		
Bachelor of Medicine	2	2
Diploma/Bachelor of Nursing	6	8
Bachelor of Pharmacy	2	3
**Work experience**		
1–5 years	3	5
6–10 years	4	7
˃10 years	3	1

## Data Availability

The data presented in this study are available on request from the corresponding author. The data are not publicly available due to restrictions set by the human research ethics committee.

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
