# Peer review of "Why Do We Not Follow Lifesaving Rules? Factors Affecting Nonadherence to COVID-19 Prevention Guidelines in Indonesia: Healthcare Professionals’ Perspectives"

_ijerph, 2022, doi:10.3390/ijerph19148502_

Round 1

Reviewer 1 Report

Hello,

Dear Authors,

What a delight to read your manuscript. I have learned a great deal and the level of detail that you put forward it absolutely wonderful. I am very impressed by your work and I have only two requests.

1. What does the abbreviation NTT stand for?

2. When was the study actually conducted?

Author Response

Thank you very much for your positive comments.

What a delight to read your manuscript. I have learned a great deal and the level of detail that you put forward it absolutely wonderful. I am very impressed by your work and I have only two requests.

  1. What does the abbreviation NTT stand for?

Response:

  • ….East Nusa Tenggara (Nusa Tenggara Timur or NTT), Indonesia.
  1. When was the study actually conducted?

Response:

  • The study was conducted in Belu and Malaka districts, NTT, Indonesia from February to March 2022.

Reviewer 2 Report

Why Don’t We Follow Lifesaving Rules? Factors Affecting 2 Nonadherence to COVID-19 Prevention Guidelines in Indone-3 sia: Healthcare Professionals’ Perspectives

After reading your proposal, I congratulate you on the work you have done and I will like to point out some of the issues that I consider important in your work.

1.              Introduction:

After reading the title of his manuscript he believed that he would know the perceptions of professionals within their work environment, of their specific task as health professionals and not as individuals in society.

Understanding that it is a qualitative study, I have perceived too many contributions that move away from the care practice and its socio-sanitary work, and too many related to their personal lives.

Why interview healthcare professionals, if they are asked about their personal life and perception of their social and non-professional life?

I believe that this contributes little to the scientific community.

Line 84-91 I do not consider that they make a significant finding to propose intervention programs with greater adherence.

2.              Methodology

I cannot understand the explanation he makes at the methodological level with his theoretical framework, and the questions or lines of subsequent analysis. Please explain better, and make sure that what you expose at the methodological level is connected with the subsequent presentation of the results.

They explain excessively the particularities of each of the chosen populations, but it is not clear why these two regions.

OBJECTIVE: The objective of this study was to investigate the challenges for adherence to COVID-19 

 preventive guidelines and practices by exploring the ecosystem of experiences and perceptions of health professionals working in healthcare delivery in Belu and Malaka districts, NTT.

After reading their goal and seeing the expected results, I do not understand why the results are based mainly on perceptions or experiences of their personal lives and not on their experience as health professionals.

3.              Results:

I recommend, to facilitate the understanding in the presentation of the results, a table where they refer to the theoretical dimensions on which they have based their questions in the interviews, and on which they must support their results.

It is not clear to me if the information is about the adherence of the professionals themselves in their workplaces, in their daily lives, the perception about the adhesion of the population they care for or treat, the adherence to their proposals as health agents ...

I believe that it must redefine its objectives and its results must be clearer to know what information they present. And thus determine whether or not it is relevant to the scientific community.

Author Response

Thank you very much for your constructive comments, our responses are attached.

Reviewer 3 Report

The title clearly and precisely reflect the content of the manuscript.

The abstract is correct. It is adequate to the issues presented in the article.

The Authors explain the aim which was to understand Indonesian healthcare professionals’ perception and experiences regarding barriers at adherence to COVID-19 prevention guidelines.

The part concerning the research procedure is compatible with the requirements. Discussion section is adequate to the research problems.

The undertaken research problem is very important in ralation to public health perspective.

Author Response

The title clearly and precisely reflect the content of the manuscript.

The abstract is correct. It is adequate to the issues presented in the article.

The Authors explain the aim which was to understand Indonesian healthcare professionals’ perception and experiences regarding barriers at adherence to COVID-19 prevention guidelines.

The part concerning the research procedure is compatible with the requirements. Discussion section is adequate to the research problems.

The undertaken research problem is very important in relation to public health perspective.

Response:

  • Thank you very much for your positive comments

Reviewer 4 Report

Thank you for the opportunity to review the work. The work is interesting and very laborious. Below are my comments:

1. I think that the text in lines 94-102 should be in the theoretical part;

2. Please expand the abbreviation NTT in the abstract where it appears for the first time;

3. Please narrow down your description in chapter 2.2. to suit 'Study setting and design'. A lot of information related to the number of population and number of hospitals does not correspond to the topic of the subchapter. Please clearly describe Study setting and design, because information from paragraph 158 is only information corresponding to chapter 2.2;

4. Please list the inclusion and exclusion criteria from the study in points;

5. In table 1, please provide the% of respondents.

Good luck to the authors and please make corrections.

Yours faithfully,

Reviewer

Author Response

Thank you very much for your positive comments.

Thank you for the opportunity to review the work. The work is interesting and very laborious. Below are my comments:

  1. I think that the text in lines 94-102 should be in the theoretical part;

Response:

  • This part has been revised and moved to introduction as recommended by reviewer 5.
  1. Please expand the abbreviation NTT in the abstract where it appears for the first time;

Response:

  • ….East Nusa Tenggara (Nusa Tenggara Timur or NTT), Indonesia.
  1. Please narrow down your description in chapter 2.2. to suit 'Study setting and design'. A lot of information related to the number of population and number of hospitals does not correspond to the topic of the subchapter. Please clearly describe Study setting and design, because information from paragraph 158 is only information corresponding to chapter 2.2;

Response:

  • Thanks for the comments, these aspects have been revised and study setting and study design have been separated into two different sub-headings.
  1. Please list the inclusion and exclusion criteria from the study in points;

Response:

  • Participants were recruited based on several inclusion criteria, including one had to be 18 years old or above, a healthcare professional and willing to voluntarily participate in the study.
  1. In table 1, please provide the% of respondents.

Response:

  • The number of respondents is to small, so we think it is not necessary to put the percentage.

Reviewer 5 Report

Thank you for the opportunity to review this work which investigates barriers to adherence to COVID-19 prevention guidelines from the perspective of healthcare professional. The qualitative design of the study necessarily limits the strength of the results presented and the significance of contents. However, the article is full of information and the authors put quite some efforts in organizing them in a fairly rational way in the 'Results' section. I appreciated the 'Limitations and strenghs' section. Nonetheless, there are some issue to address, especially in the 'Methods' part:

Some of 2.1 and 2.2 belongs more to 'Introduction' than to 'Methods', for example: lines 93-102, and 125-130;

Table 1 should be moved to the 'Results' section;

There are typos to be corrected in lines 206-207;

Line 467-469: unclear sentence.

'Conclusion' must be shortened. Minor typos and language spells to be checked and corrected.

Author Response

Thank you very much for your positive comments.

Thank you for the opportunity to review this work which investigates barriers to adherence to COVID-19 prevention guidelines from the perspective of healthcare professional. The qualitative design of the study necessarily limits the strength of the results presented and the significance of contents. However, the article is full of information and the authors put quite some efforts in organizing them in a fairly rational way in the 'Results' section. I appreciated the 'Limitations and strenghs' section. Nonetheless, there are some issue to address, especially in the 'Methods' part:

Some of 2.1 and 2.2 belongs more to 'Introduction' than to 'Methods', for example: lines 93-102, and 125-130;

Response:

  • Thank you very much for this detailed observation:
    • This has been revised
    • Lines 93-102 have been integrated into introduction section

Table 1 should be moved to the 'Results' section;

Response:

  • As we described about the participants in the methods section, so we think Table 1 which contains sociodemographic profile of the participants is proper to place it within the methods section.

There are typos to be corrected in lines 206-207;

Response:

  • These have been revised:

They were also advised that ethics approval for this study was obtained from the Health Research Ethics Committee, Duta Wacana Christian University, Indonesia (No. 1380/C.16/FK/2022).

Line 467-469: unclear sentence.

Response:

  • Revision has been done:

Guided by TPB concepts [24], our analysis demonstrates two sets of perceived standards operating in the community as reported by HCPs which led to nonadherence to COVID-19 preventive behaviours. HCPs noncompliance resulted from social pressure despite having extensive biomedical knowledge regarding the pandemic, while they reported that community members’ nonadherence resulted from social pressure because of a lack of biomedical awareness, ignorance, etc.

'Conclusion' must be shortened. Minor typos and language spells to be checked and corrected.

Response:

  • Conclusion has been shortened

The COVID-19 pandemic did not occur in a social vacuum; as it made its way through Indonesia, it encountered traditions, cultures, and preconceptions influencing how disease entities impact communities and health systems. This study confirmed that personal behavioural outcomes are influenced by intention, attitude, subjective norms, and perceived behavioural control, and that the various confluence of these factors can determine how an individual will behave. Structural constraints such as poverty and lack of material resources had an impact on adherence to preventive behaviours, as reported elsewhere in the literature. The strongest determinant for nonadherence, as reported by HCPs was the need for compliance with subjective social norms in social events, in public, and at gatherings. The pressure to appear ‘normal’ and reject the use of masks or physical distancing was more influential than personal knowledge, access to resources, or others.

The findings indicate that policymakers in remote, multicultural locales in Indonesia such as NTT must take into consideration that familial and traditional (social) ties and bonds override individual agency where personal action is strongly guided by long-held social norms. The impetus for belonging and acceptance, and to maintain psychological wellbeing is inextricably linked to social performance, cohesion, and participation in community-endorsed rituals and activities. Thus, while agency-focused preventive policies which encourage individual actions (hand washing, mask wearing) are essential, in NTT they must be augmented by social change, advocating with trusted traditional (adat) and religious leaders to revise norms in the context of a highly transmissible pandemic virus. This study demonstrates that trust and social cohesion are the primary drivers and determinants for adherence to individual prevention, and must be considered in future policy design.

Round 2

Reviewer 2 Report

After reading the modifications to the recommendations presented in the first phase of the review, considering that the work continues to have flaws that are difficult to modify, since they are part of the development of the study and its objective.

Author Response

Thanks for your comments. As you previously suggested to make clear whether the information is about the adherence of the professionals themselves in their workplaces or in their daily lives, or the perception about the adherence of the population they care for or treat.

We have made it clear that information reported is about sociocultural factors that impede adherence of HCPs and community members COVID-19 prevention protocols in their social life. These are reflected in the main interview questions provided in Table 1 and the results section.

The highly decentralised and locally-managed regulatory environment of the Indonesian COVID-19 response requires unique, tailor-made, and evidence-based approaches regarding community outreach, preventive strategies, and management of COVID-19 policy and guidelines. As there are limitations in the scholarly literature on the sociocultural barriers and constraints of HCP and community adherence to preventive guidelines globally and in the multicultural context of Eastern Indonesia, coupled with limitations in outreach, communication, and enforcement resources, we have chosen to investigate perspectives of HCPs about constraints in preventing behaviours among them and community members in their social life in Belu and Malaka districts, East Nusa Tenggara (Nusa Tenggara Timur or NTT), Indonesia. While an individual’s behaviour based on personal agency is not the sole determinant of pandemic outcomes, positive or negative, the social disruptions seen during COVID-19 necessitate a full analysis of individual, social, and structural determinants of health outcomes. This study contributes to improving our understanding of sociocultural barriers and the individual psychological context of pandemic prevention behaviours as important components of a comprehensive, resilient strategy for future pandemic control. Understanding barriers and constraints of adherence is useful to inform the development of policies and intervention programs to support people’s adherence to the guidelines and improve COVID-19 related healthcare service delivery.

Reviewer 5 Report

The authors have improved the manuscript in a satisfactory way.

Author Response

Dear reviewer,

Thank you very much.